# Modification of Regulatory T Cell Epitopes Promotes Effector T Cell Responses to Aspartyl/Asparaginyl β-Hydroxylase

**DOI:** 10.3390/ijms232012444

**Published:** 2022-10-18

**Authors:** Sebastian Wirsching, Michael Fichter, Maximiliano L. Cacicedo, Katharina Landfester, Stephan Gehring

**Affiliations:** 1Children’s Hospital, University Medical Center of the Johannes Gutenberg University Mainz, Langenbeckstr. 1, 55131 Mainz, Germany; 2Department of Dermatology, University Medical Center of the Johannes Gutenberg University Mainz, Langenbeckstr. 1, 55131 Mainz, Germany; 3Max Planck Institute for Polymer Research, Ackermannweg 10, 55128 Mainz, Germany

**Keywords:** cancer, immunotherapy, T cell activation, protein modification, regulatory T cell epitopes

## Abstract

Cancer is a leading cause of death worldwide. The search for innovative therapeutic approaches is a principal focus of medical research. Vaccine strategies targeting a number of tumor-associated antigens are currently being evaluated. To date, none have garnered significant success. Purportedly, an immunosuppressive tumor microenvironment and the accumulation of regulatory T cells contribute to a lack of tumor vaccine efficacy. Aspartyl/asparaginyl β-hydroxylase (ASPH), a promising therapeutic target, is overexpressed in a variety of malignant tumors but is expressed negligibly in normal tissues. Computer analysis predicted that ASPH expresses four peptide sequences (epitopes) capable of stimulating regulatory T cell activity. The abolition of these putative regulatory T cell epitopes increased the CD4^+^ and CD8^+^ effector T cell responses to monocyte-derived dendritic cells pulsed with a modified, epitope-depleted version of ASPH in an ex vivo human lymphoid tissue-equivalent coculture system while simultaneously decreasing the overall number of FoxP3^+^ regulatory T cells. These findings suggest that the efficacy of all new vaccine candidates would profit from screening and eliminating potential tolerogenic regulatory T cell epitopes.

## 1. Introduction

Cancer is a major cause of death worldwide; approximately 19.3 M new cases and 10 M deaths occurred in 2020 [1]. Metastasis is the principal cause. Identifying innovative therapeutic approaches targeting the factors that affect malignant cell migration and invasion is critical.

Aspartyl/asparaginyl β-hydroxylase (ASPH) is a Type II transmembrane protein that belongs to the α-ketoglutarate-dependent dioxygenase family of prolyl and lysyl hydroxylases [2]. It is a highly conserved enzyme that catalyzes the hydroxylation of aspartyl and asparaginyl residues in the epidermal growth factor-like domains found in target proteins, e.g., Notch receptors and ligands implicated in malignant transformation [3,4]. ASPH is overexpressed in a wide variety of malignant tumors compared to normal human tissues [4,5]. ASPH overexpression—in the case of hepatocellular carcinoma (HCC), for example—produces a malignant phenotype characterized by increased cell motility, invasion and metastases [6]. Moreover, the level of ASPH expression in HCC correlates inversely with postoperative prognosis and patient survival [4,7]. The elevated expression of ASPH on the surface of HCC tumor cells but not cells in surrounding, noninvolved tissues suggests that ASPH could serve as an immunotherapeutic target [8]. Indeed, immunization with ASPH-loaded dendritic cells or ASPH-expressing λ phage constructs suppressed tumor cell growth in a mouse model of HCC; both CD4^+^ and CD8^+^ effector T_(eff)_ cells contributed to tumor suppression [9,10].

The liver is inherently tolerogenic, an environment exacerbated by chronic inflammation and developing neoplasia in HCC patients [11,12]. An elevated FoxP3^+^ regulatory T_(reg)_ cell population observed in HCC cases contributes to a weakened, ineffective tumor-specific T cell response [13,14]. It is relevant, therefore, that eliminating T_reg_ cells prior to incubating pan T cells and ASPH protein-pulsed monocyte-derived dendritic cells (moDCs) derived from HCC patients significantly increased the percentage of activated CD4^+^ and CD8^+^ T cells recovered in an ex vivo, human lymphoid tissue equivalent culture system [15]. This finding suggests that the overexpression of ASPH in the livers of HCC patients might promote T_reg_ cell function and immunosuppression. Furthermore, it implies that ASPH includes one or more peptide sequences (epitopes) that are capable of stimulating T_reg_ cell activity.

Computer analysis enables the identification of epitopes that are expressed by a large number of proteins and are cross-conserved at the T cell receptor (TcR) facing aspect; these epitopes are predicted to induce T_reg_ cell activity [16]. Four peptide sequences identified by the analysis of ASPH are cross-conserved at the TcR face with epitopes associated with between 13 and 31 other human proteins. Here, the modification of these four epitopes, which prevented MHC binding and presentation, enhanced both the CD4^+^ and CD8^+^ T cell responses and diminished the T_reg_ cell response to ASPH (modified) in an ex vivo, human lymphoid tissue equivalent culture system. These findings indicate that any strategy for using ASPH as a therapeutic vaccine in cancer treatment would benefit tremendously from modifying these four sequences.

## 2. Results

### 2.1. Recombinant ASPH and ASPHmod

The wild-type ASPH sequence was modified by changing four 9-mers that were predicted to comprise epitopes that are cross-conserved at the TcR face with epitopes associated with numerous other human proteins and, consequently, recognized by T_reg_ cells [16]. Beside these epitope sequences, ASPH and ASPHmod were identical. Both sequences were altered by introducing a tissue plasminogen activator (tPA) signal peptide sequence and deleting the transmembrane domain (Figure 1). 

The identity and purity of both of the recombinant proteins were determined (Figure 2). Western blot analysis revealed three ASPH fragments, which ranged in size from approximately 120 kDA to 70 kDA. The analysis of ASPHmod, on the other hand, showed a single fragment that was approximately 120 kDA in size. The purity of both protein fractions, assessed by silver staining, was >90%.

### 2.2. Elevated T Cell Responses to ASPHmod

The immunostimulatory activities of ASPH and ASPHmod were compared in an ex vivo moDC:T cell coculture system conducted in accordance with methods originally described by others [17]. Priming and then restimulating T cells with ASPHmod-pulsed moDCs increased the number of activated CD4^+^CD154^+^IFN-γ^+^ cells relative to the number of those induced by moDCs pulsed with non-modified ASPH (Figure 3A,B). The combined data derived from nine individuals confirmed this increased response to ASPHmod. T cells restimulated with ASPHmod exhibited 4.7-fold more reactive, triple-positive T cells relative to the non-pulsed, negative control (Figure 3C). ASPH-stimulated T cells, on the other hand, showed only a marginal increase. Moreover, a comparison of the T cell responses of the same nine individuals cocultured with ASPH- and ASPHmod-moDCs uniformly showed a greater response to the latter (Appendix A).

Similarly, a significantly greater number of TNF-α-producing T cells (CD4^+^CD154^+^TNF-α^+^) was generated in cocultures that contained ASPHmod-pulsed, versus ASPH-pulsed, moDCs (Figure 4A,B). An average 2.9-fold increase in the number of CD4^+^CD154^+^TNF-α^+^ T cells derived from cocultures pulsed with ASPHmod relative to non-pulsed cocultures was observed; only a negligible increase in triple-positive T cells was found in cocultures pulsed with ASPH (Figure 4C). Moreover, the fold change in the number of reactive T cells, relative to the non-pulsed control, was consistently greater when comparing the responses of the same individuals to ASPHmod and ASPH (Figure 4D).

ASPHmod-pulsed moDCs also stimulated a greater CD8^+^ T cell response than ASPH-pulsed or non-pulsed moDCs. The number of CD8^+^IFN-γ^+^ T cells was significantly higher than the number of IFN-γ^+^ cells derived from ASPH-pulsed or non-pulsed cocultures (Figure 5A,B). The data obtained from multiple leukocyte donors demonstrated a significantly greater fold change in CD8^+^IFN-γ^+^ T cells generated in cocultures that contained ASPHmod-pulsed moDCs (Figure 5C). Relative to non-pulsed moDCs, each donors exhibited a greater response to ASPHmod-pulsed moDCs than they did to ASPH-pulsed moDCs (Figure 5D). An increased, albeit statistically insignificant, number of TNF-α-producing CD8^+^ T cells was also found among cells obtained from cocultures composed of ASPHmod-pulsed moDCs compared to ASPH-pulsed moDCs (Appendix A).

The response of the T cells cocultured with either ASPH-pulsed or ASPHmod-pulsed moDCs was also quantified and compared in terms of cell proliferation and cytokine production by restimulating sensitized T cells with non-pulsed or antigen-pulsed moDCs in a second experiment, separate from ICCS. Sensitized T cells restimulated with ASPHmod-pulsed moDCs exhibited a significant increase in proliferating cells compared to cells restimulated with ASPH-pulsed moDCs (Figure 6A). The latter failed to proliferate more than the negative control, i.e., T cells incubated with non-pulsed moDCs. Upon restimulation, T cells derived from ASPHmod-pulsed DC cocultures secreted more IFN-γ, TNF-α and IL-2, as assessed by the cytometric bead array (Figure 6B–D). In contrast, T cells cocultured with ASPH-pulsed moDCs secreted approximately the same amount of cytokines as the cells cocultured under control conditions.

Moreover, pulsing moDCs with ASPHmod in combination with the four wild type, regulatory epitopes abrogated the increased immunogenicity of ASPHmod in both CD4^+^ and CD8^+^ T cell populations after coculture (Appendix A).

### 2.3. Increased ASPHmod Immunogenicity Correlates Inversely with T_reg_ Cell Number

Experiments were undertaken to determine the relationship between the modification of the four predicted T_reg_ cell epitopes in ASPH (creating ASPHmod), T_reg_ cell activity and the enhanced T_eff_ cell responses to ASPHmod-pulsed moDCs determined in cocultures. Phenotyping T cells collected after incubating naïve T cells derived from a single donor for 14 days revealed fewer CD4^+^FoxP3^+^ T cells in ASPHmod-pulsed cocultures compared to ASPH-pulsed or non-pulsed cocultures (Figure 7A,B). An analysis of the combined data obtained from five individuals yielded the same result—a 20% reduction, on average, in the number of FoxP3^+^ T cells recovered from cocultures that contained ASPHmod-pulsed, compared to ASPH-pulsed, moDCs (Figure 7C,D).

## 3. Discussion

Cancer is a major cause of death globally. The search for novel treatments is a principal focus of medical research. One approach, the development of immunotherapeutic vaccines, is supported by the observation that the recurrence of disease following treatment, e.g., tumor resection or organ transplant in cases of HCC, is reduced in patients with tumors that contain infiltrating (presumably tumor-specific) T_eff_ cells [18]. The elevated, tumor-specific response of CD4^+^ and CD8^+^ T cells in HCC patients following ablation and the spontaneous regression of untreated (distant) as well as ablated primary tumors in animal models suggest that ablation elicits a systemic, albeit not curative, anti-tumor response [19,20]. As such, maximizing anti-tumor immunity should enhance therapy and diminish tumor recurrence.

A number of cancer vaccine strategies are being evaluated: cell-based (tumor or immune cell), genetic-based (RNA, DNA or viral) and protein- or peptide-based [21]. To be effective, therapeutic vaccines must target antigen(s) to DCs, induce DC activation and stimulate robust CD4^+^ and CD8^+^ T_eff_ cell responses; antigenic choice is paramount [22,23]. A variety of tumor-associated antigens (TAAs) have been used in efforts to develop therapeutic vaccines against non-viral cancers: NY-ESO-1, MAGE-A3, BAGE, CEA, AFP, XAGE-1B, survivin, P531, h-TERT, mesothelin, PRAME and MUC-1 [22]. To date, none of the vaccine strategies that target these TAAs have garnered significant success in clinical trials; additional/novel TAAs (e.g., ASPH) need to be evaluated [21,24]. Relative to normal human tissues, ASPH is overexpressed in a wide variety of malignant tumors affecting the pancreas, breast, prostate, cervix, ovary, fallopian tube, larynx, lung, thyroid, gall bladder, kidney, bladder, brain, gastrointestinal tract (esophagus, stomach and colon) and liver (cholangiocellular carcinoma, HCC) and includes malignancies that commonly affect children (i.e., glioblastoma and Wilms’ tumor) [3,5].

It seems paradoxical that TAA-specific CD8^+^ T cells and tumor progression co-exist in patients with advanced cancer [25]. Purportedly, an immunosuppressive tumor microenvironment (TME) contributes significantly to CD8^+^ T cell exhaustion and a lack of anti-tumor vaccine efficacy. Multiple immunoregulatory pathways impede T_eff_ cell–mediated tumor destruction in the TME [22,25]. Exhausted CD8^+^ T cells progressively lose their capacities to proliferate, produce cytokines and lyse antigen-expressing target cells; exhaustion correlates with low CD4^+^ T helper cell activity [25]. The loss of CD8^+^ T_eff_ cell function occurs concomitantly with the increased cell-surface expression of multiple inhibitory checkpoint receptors including program death (PD)-1, cytotoxic T lymphocyte antigen (CTLA)-4, T cell immunoglobulin, mucin (Tim)-3 and lymphocyte activation gene (LAG)-3, among others [25]. In addition, T_reg_ cells accumulated in the TME exert an immunosuppressive effect on T_eff_ cell function by inhibiting DC maturation and a productive interaction with T_eff_ cells [14,25]. T_reg_ cells also produce immunosuppressive factors, e.g., IL-10, TGF-β and adenosine, which contribute to T_eff_ cell exhaustion.

The response of CD4^+^ and CD8^+^ T cells to ASPH in an ex vivo coculture system analogous to the one described herein increased significantly after the removal of CD25^+^ T_reg_ cells prior to the culture [15]. Computer analysis predicted the presence of four peptide sequences (epitopes) in ASPH capable of stimulating T_reg_ cell activity and immunosuppression [16,26]. These four sequences were modified in the present study, eliminating their ability to bind class II MHC molecules and, thus, stimulate T_reg_ cell activity. These modifications resulted in a protein product that was synthesized as a single fragment rather than three fragments of varying sizes, characteristic of unmodified ASPH. The results of SDS-PAGE were similar to those reported by Shimoda et al., who synthesized recombinant ASPH in a baculovirus system [9].

The computer analysis and the ASPH sequences predicted to stimulate T_reg_ cell activity were validated in an ex vivo, human lymphoid tissue equivalent culture system. Naïve T cells sensitized and then restimulated by culture with ASPHmod-pulsed moDCs exhibited a far greater response than T cells cultured with moDCs pulsed with unmodified ASPH. CD4^+^ T cells, induced and stimulated with ASPHmod-pulsed DCs, exhibited a significantly greater number of CD154^+^IFN-γ^+^ and CD154^+^TNF-α^+^ cells; the number of CD8^+^IFN-γ^+^ T cells was also increased substantially. Notably, naïve T cells derived from the same individual always exhibited a greater response to ASPHmod-pulsed, versus ASPH-pulsed, moDCs. Indeed, the response to ASPH-pulsed moDCs was often less than the response to non-pulsed DCs.

Sensitized T cells cocultured with ASPHmod-pulsed DCs also proliferated to a greater extent and secreted substantially more IL-2, IFN-γ and TNF-α than cells incubated with moDCs pulsed with ASPH. Furthermore, spiking in the four wild type, regulatory epitopes abrogated the increased immunogenicity against ASPHmod. Importantly, the elimination of the predicted T_reg_ cell epitopes in ASPHmod resulted in a reduction in the number of FoxP3^+^ T_reg_ cells generated in cocultures containing ASPHmod-pulsed, versus ASPH-pulsed, moDCs. 

Recently, Iwagami et al. reported the therapeutic efficacy of two λ phage-based vaccine constructs that display ASPH protein sequences bound to the gpD protein located on the phage surface [10]. The λ1 construct displayed the N-terminal third of human ASPH; λ3 displayed the C-terminal third. Mice vaccinated prophylactically with either construct exhibited a marked growth reduction in ASPH-expressing tumor cells inoculated subcutaneously and an increased rate of survival. Cultured splenocytes derived from ASPH-λ3-vaccinated animals also exhibited marked increases in ASPH-specific CD4^+^ and CD8^+^ T cells and the production of TH2-type cytokines (i.e., IL-4, IL-6 and IL-10). On the other hand, the production of the TH1-type cytokines (i.e., IFN-γ and TNF-α) was only marginally increased—or sharply reduced in the case of IL-2—relative to the production by splenocytes derived from non-vaccinated animals. Notably, both ASPH-λ1 and ASPH-λ3 express the putative T_reg_ cell epitopes described here, which are capable of eliciting T_reg_ cell activity and IL-10 production while suppressing the synthesis of TH1-type cytokines [14,27]. These findings suggest that strategies for using ASPH-λ1 (PAN-301-1) to treat prostate cancer patients (ClinicalTrials.gov Identifier: NCT03120832) or patients with squamous cell carcinoma of the head and neck (ClinicalTrials.gov Identifier: NCT04034225) would benefit from the abolition of these putative T_reg_ cell epitopes [28,29].

In summary, ASPH is overexpressed in a wide variety of malignant tumors, but it is not expressed in an appreciable amount in normal adult human tissue. As such, it represents a promising target for immunotherapy [30]. The results presented herein indicate that the unmodified ASPH protein contains four sequences (epitopes) capable of stimulating T_reg_ cell function and, as a consequence, suppressing T_eff_ cell proliferation and activity. Conceivably, the ability of ASPH to promote T_reg_ cell function contributes to the immunosuppressive microenvironment commonly found in tumors. The abolition of these putative T_reg_ cell epitopes increased the CD4^+^ and CD8^+^ T cell responses to ASPHmod in an ex vivo coculture system while simultaneously decreasing the overall number of FoxP3^+^ T_reg_ cells. These findings indicate that the efficacy of all new vaccine candidates would profit significantly from screening and subsequently eliminating potential tolerogenic T_reg_ cell epitopes prior to entering clinical trials.

## 4. Materials and Methods

### 4.1. Recombinant ASPH and Modified ASPH (ASPHmod)

Full length ASPH (GenBank Accession number S83325) was altered by introducing a tissue plasminogen activator (tPA) signal peptide sequence at the beginning and a histidine-tag at the end. The transmembrane domain (position 54–74) was also deleted. A computer analysis of ASPH performed by EpiVax, Inc. (Providence, RI) indicated that four peptide sequences—VLLGLKERS (position 104–112), VKKKKPKLL (position 325–333), DLLKLSLKR (position 416–424) and WHPELTPQQ (position 742–750)—were cross-conserved at the TcR interface with sequences found in 31, 24, 18 and 13 other human proteins, respectively. Highly cross-conserved sequences are predicted to encode T_reg_ cell epitopes and induce T_reg_ cell activity [16]. ASPH was modified to eliminate these epitopes by substituting the amino acids at positions 1, 4, 6 and 9 with alanine, creating ASPHmod.

Both ASPH and ASPHmod were cloned into the NotI site of the gWIZ vector (Genlantis Inc., San Diego, CA, USA). After cloning, the constructs were sequenced using a Mix2Seq kit (Eurofins Genomics) in order to verify both sequences. Recombinant proteins were synthesized using the Expi293^TM^ Expression System (ThermoFisher Scientific, Waltham, MA, USA) and purified by passage over HisTrap HP columns (GE Healthcare, Chicago, IL). Recombinant proteins were subjected to western blot analysis using the anti-ASPH antibody (Invitrogen, Waltham, MA, USA). Recombinant protein purity was evaluated by Silver Stain (ThermoFisher Scientific).

### 4.2. Ex Vivo, Human Lymphoid Tissue-Equivalent Culture System

Leukocytes were obtained from leukapheresis products collected from healthy, voluntary donors (blood bank of the University Medical Center Mainz) upon informed consent. ASPH- and ASPHmod-specific T cells were generated ex vivo in accordance with the methods described by Moser et al. [17]. Peripheral blood mononuclear cells (PBMCs) were isolated by density gradient centrifugation on Histopaque-1077 (Sigma-Aldrich, Steinheim am Albuch, Germany). Two-thirds of the PBMCs were suspended in Cryo-SFM medium (PromoCell, Heidelberg, Germany), frozen and stored in liquid nitrogen for later use. The monocytes were isolated from the remaining PBMCs using a CD14 Microbeads kit purchased from Miltenyi Biotec (Bergisch Gladbach, Germany). Six-well suspension culture plates (Greiner Bio-One, Kremsmunster, Austria) were inoculated with purified monocytes in X-VIVO 15 medium (Lonza, Walkerville, MD) supplemented with 1% penicillin-streptomycin, 100 ng/mL GM-CSF (ImmunoTools GmbH, Friesoythe, Germany) and 25 ng/mL IL-4 (ImmunoTools) and incubated at 37 °C and 5% CO_2_. Half of the spent medium was replenished with fresh X-VIVO 15 medium supplemented with cytokines after 3 days. The cells were either non-pulsed (negative control) or pulsed with 1 µg/mL of ASPH or ASPHmod after an additional 2-day incubation. The cells were incubated for another 24 h and then induced to mature by adding 25 ng/mL TNF-α. Mature, antigen-pulsed moDCs were collected after 48 h.

ASPH- and ASPHmod-specific T cells were produced from autologous T cells isolated from frozen PBMCs using a Pan T Cell Isolation kit (Miltenyi Biotec). Isolated T cells (2.1 × 10^6^) were seeded into 24-well cell culture plates and cocultured for 14 days with 3.5 × 10^4^ mature non-pulsed, ASPH-pulsed or ASPHmod-pulsed moDCs in X-VIVO 15 medium supplemented with 1% penicillin-streptomycin. Half of the spent medium was replaced with fresh medium every 4–5 days. At the end of the 14-day incubation period, the antigen-sensitized T cells were collected and restimulated with fresh non-pulsed, ASPH-pulsed or ASPHmod-pulsed moDCs that were generated as described in the preceding paragraph. T cell activation was quantified by flow cytometric analysis.

### 4.3. Intracellular Cytokine Staining (ICCS) and Flow Cytometric Analysis

Antigen-specific T cells (2 × 10^5^) were seeded in triplicate into 96-well round-bottom plates and stimulated with 5 × 10^4^ non-pulsed, ASPH-pulsed or ASPHmod-pulsed moDCs. The cells were incubated for 7 h at 37 °C and 5% CO_2_. Then, 5 µg/mL Brefeldin A (BioLegend, San Diego, CA) was added to each well after two hours to block secretion. The cells were collected at the end of the 7 h incubation period, transferred to FACS tubes and washed. ICCS and flow cytometry were performed in accordance with the methods we described previously [31,32]. Briefly, activated ASPH and ASPHmod T lymphocytes in FACS tubes were incubated with 10% Privigen^®^ Immunoglobulin solution (CSL Behring, Marburg, Germany) to block Fc receptors. Subsequently, the cells were stained extracellularly with fluorophore-conjugated mouse monoclonal antibodies specific for CD4 (FITC) and CD8 (BV510) (BD Biosciences; Franklin, NJ, USA). The cells were then washed and permeabilized using the Cytofix/Cytoperm kit (BD Biosciences) according to the manufacturer’s instructions. The permeabilized cells were stained intracellularly with fluorophore-conjugated monoclonal antibodies specific for CD154 (PE), IFN-γ (PE-Cy7) and TNF-α (APC) (BD Biosciences). The stained cells were quantified using an LSR II (BD Biosciences) or a MACSQuant^®^ Analyzer 16 flow cytometer (Miltenyi Biotec). The flow cytometric data were evaluated using FlowLogic software version 8.4 (Inivai Technologies, Victoria 3194 Australia) or FlowJo software version 10.8 (Ashland, OR: Becton, Dickinson and Company).

To enumerate T_reg_ cells, 5 × 10^5^ sensitized T cells collected after the initial 14-day coculture period were transferred to FACS tubes and stained extracellularly with fluorophore-conjugated mouse monoclonal antibodies specific for CD3 (FITC), CD4 (APC) and CD8 (BV510) (BD Biosciences). Subsequently, the stained cells were washed and permeabilized using the FoxP3 buffer kit (BD Biosciences). The permeabilized cells were then stained intracellularly with fluorophore-conjugated monoclonal antibody specific for FoxP3 (PE) (BD Biosciences). The stained cells were quantified and evaluated according to the description in the preceding paragraph.

### 4.4. T Cell Proliferation

T cell proliferation was assessed in a second experiment, separate from ICCS, by coculturing 5 × 10^4^ antigen-sensitized T cells in X-VIVO 15 medium and 1% penicillin-streptomycin with 5 × 10^3^ fresh non-pulsed, ASPH-pulsed or ASPHmod-pulsed moDCs in triplicate in 96-well round-bottom plates. Following 4 days of incubation, the cells were collected and transferred to FACS tubes. Proliferation was assessed using the Cell Proliferation EdU-488 Kit III (PromoCell, Heidelberg, Germany) according to the manufacturer’s instructions. The stained cells were quantified using an LSR II (BD Biosciences) or a MACSQuant^®^ Analyzer 16 flow cytometer (Miltenyi Biotec). The flow cytometric data were evaluated using FlowLogic software version 8.4 or FlowJo software version 10.8. Representative dot plots are included in the Appendix A.

### 4.5. Cytometric Bead Array

Cell culture supernatants, collected after 24 h of restimulation, were analyzed using the Legendplex™ Human Essential Immune Response Panel (BioLegend) according to the manufacturer’s instructions. The supernatants were quantified using an LSR II flow cytometer (BD Biosciences). The data were evaluated using the Legendplex™ Data Analysis Software Suite (BioLegend).

### 4.6. Statistical Analysis

The statistical analyses were performed using GraphPad Prism version 7 (GraphPad Software, San Diego, CA, USA). The data were initially analyzed for a Gaussian distribution using a Shapiro–Wilk normality test. Subsequently, the data were analyzed using an unpaired Student’s *t* test if the normality tests were passed; if not, a Mann–Whitney test was used to compare the fold changes between the ASPH and ASPHmod conditions. To compare the T cell responses after restimulation with non-pulsed, ASPH-pulsed or ASPHmod-pulsed moDCs, a one-way ANOVA was used if the normality tests were passed; if not, a Kruskal–Wallis test was used. *p* values < 0.05 were considered statistically significant.

## Figures and Tables

**Figure 1 ijms-23-12444-f001:**
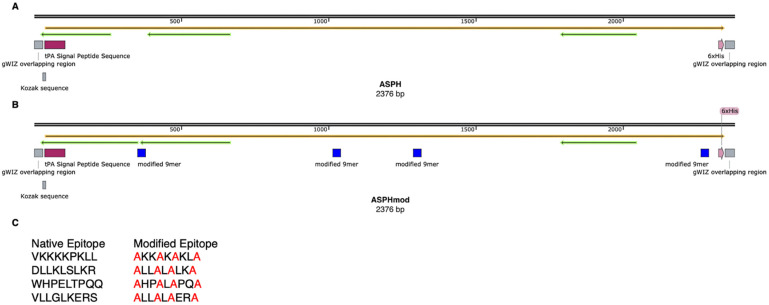
Graphic illustration of ASPH and ASPHmod. (**A**) ASPH and (**B**) ASPHmod are depicted. (**C**) Native 9-mer sequences present in ASPH and modified sequences present in ASPHmod. The illustration was prepared using SnapGene Viewer (Insightful Science; snapgene.com, accessed on 25 April 2022).

**Figure 2 ijms-23-12444-f002:**
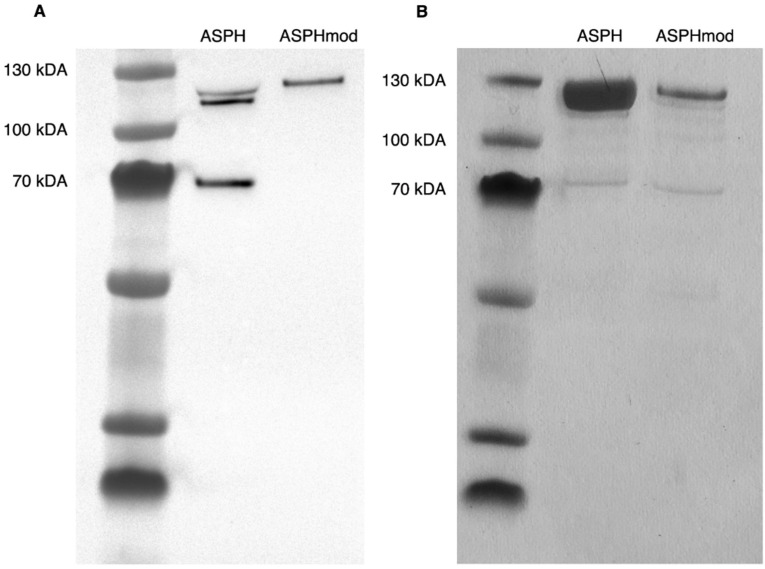
Analysis of recombinant ASPH and ASPHmod. SDS-polyacrylamide electrophoresis of recombinant ASPH and ASPHmod proteins. (**A**) Western blot analysis. (**B**) Silver stain. PageRuler^TM^ Plus Prestained Protein Ladder (5 µL; ThermoFisher Scientific, Waltham, MA, USA) provides the molecular weight standard.

**Figure 3 ijms-23-12444-f003:**
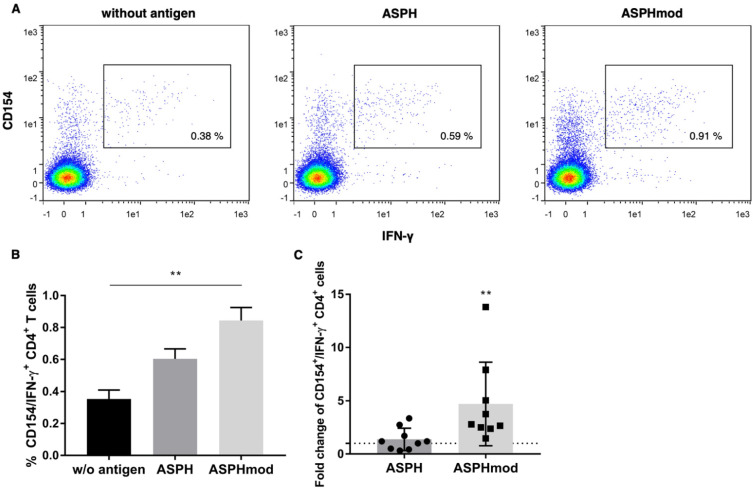
ASPHmod-pulsed moDCs induce an increase in CD4^+^CD154^+^IFN-γ^+^ T cells. (**A**) Representative flow cytometric analysis of the T cell response of a single individual. (**B**) Data represent the means ± SD of the same individual assessed in quintuplicate: ** significantly different, *p* < 0.01 (Kruskal–Wallis test). (**C**) Fold change in CD4^+^CD154^+^IFN-γ^+^ T cells relative to cells restimulated with the non-pulsed moDC control. Data represent the means ± SD of nine individuals: ** significantly greater, *p* < 0.01 (Mann–Whitney test).

**Figure 4 ijms-23-12444-f004:**
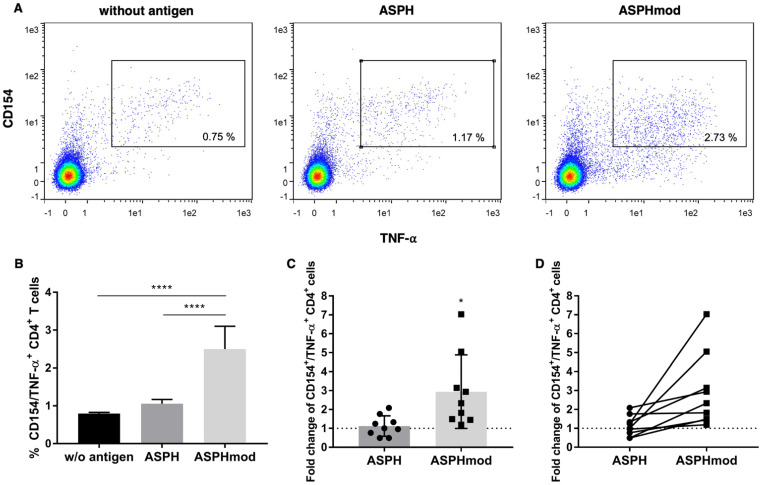
ASPHmod-pulsed moDCs induce an increase in CD4^+^CD154^+^TNF-α^+^ T cells. (**A**) Representative flow cytometric analysis of a single individual. (**B**) Data represent the means ± SD of the same individual assessed in quintuplicate: **** significantly different, *p* < 0.001 (one-way ANOVA). (**C**) Fold change in CD4^+^CD154^+^TNF-α^+^ T cells relative to cells restimulated with the non-pulsed moDC control. Data represent the means ± SD of nine individuals: * significantly greater, *p* < 0.05 (unpaired Student’s *t* test). (**D**) Relative T cell response of each individual to ASPH- versus ASPHmod-pulsed moDCs.

**Figure 5 ijms-23-12444-f005:**
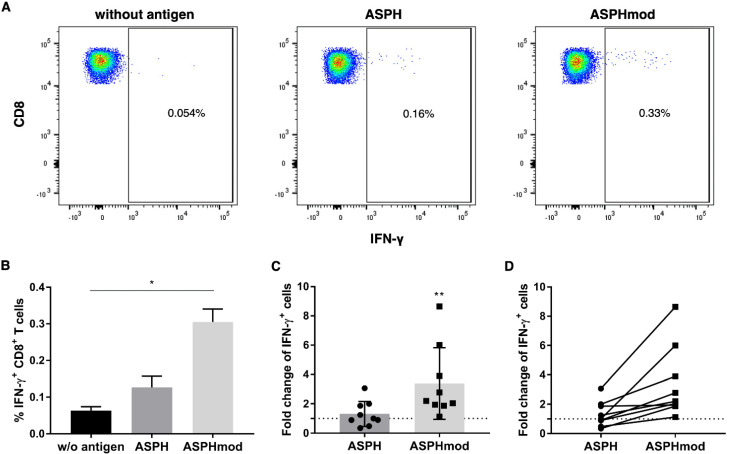
ASPHmod-pulsed moDCs stimulate an increased number of CD8^+^IFN-γ^+^ T cells. (**A**) Representative flow cytometric analysis of a single individual. (**B**) Data represent the means ± SD of the same individual assessed in triplicate: * significantly different, *p* < 0.05 (Kruskal–Wallis test). (**C**) Fold change in CD8^+^IFN-γ^+^ T cells relative to cells restimulated with the non-pulsed moDC control. Data represent the means ± SD of nine individuals: ** significantly greater, *p* < 0.01 (Mann–Whitney test). (**D**) Relative T cell response of each individual to ASPH- versus ASPHmod-pulsed moDCs.

**Figure 6 ijms-23-12444-f006:**
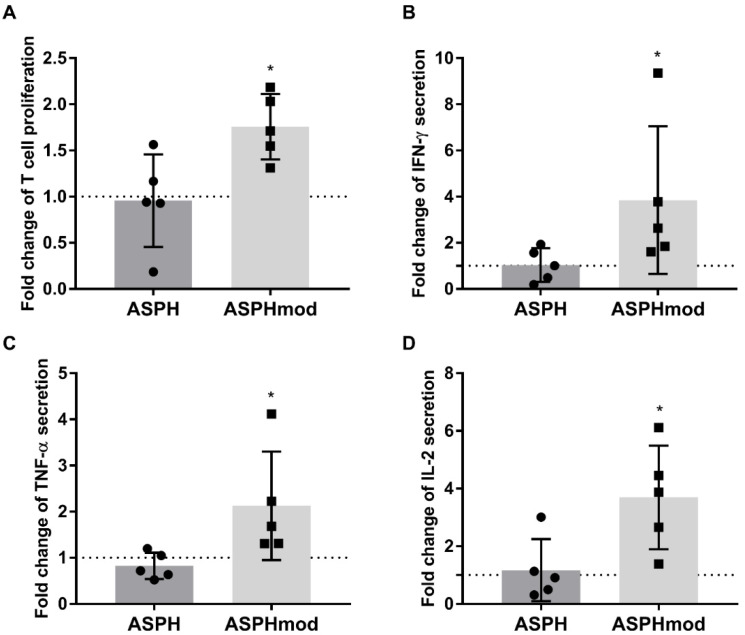
ASPHmod-pulsed moDCs stimulate increased T cell proliferation and cytokine production. Fold changes in (**A**) T cell proliferation, or the production of (**B**) IFNγ, (**C**) TNFα and (**D**) IL-2 after restimulation with ASPH- or ASPHmod-pulsed moDCs relative to cells restimulated with the non-pulsed moDC control. Data represent the means ± SD of five individuals: * significantly greater, *p* < 0.05 ((**A**,**C**,**D**): unpaired Student’s *t* test; (**B**): Mann–Whitney test).

**Figure 7 ijms-23-12444-f007:**
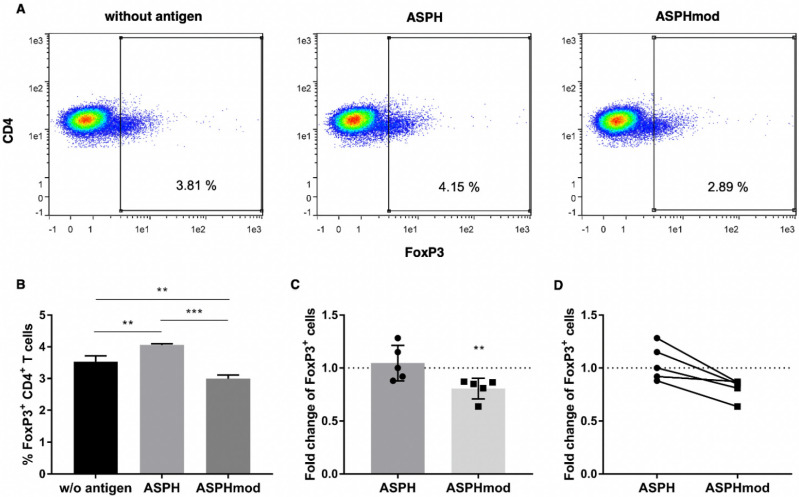
ASPHmod-pulsed moDCs induce fewer T_reg_ cells. (**A**) Representative flow cytometric analysis of a single individual. (**B**) Number of CD4^+^FoxP3^+^ T cells after restimulation with no antigen-, ASPH- or ASPHmod-pulsed DCs. Data represent the mean ± SD of a single individual donor assessed in triplicate: ** significantly different, *p* < 0.01, *** significantly different, *p* < 0.005 (one-way ANOVA). (**C**) Fold change in CD4^+^FoxP3^+^ T cells relative to cells restimulated with the non-pulsed moDC control. Data represent the means ± SD of five individuals: ** significantly less, *p* < 0.01 (Mann–Whitney test). (**D**) Relative T cell response of nine individuals to ASPH- versus ASPHmod-pulsed moDCs.

## Data Availability

The datasets generated and/or analyzed during the current study are available from the corresponding author on reasonable request.

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
