# Peer review of "Modification of Regulatory T Cell Epitopes Promotes Effector T Cell Responses to Aspartyl/Asparaginyl β-Hydroxylase"

_ijms, 2022, doi:10.3390/ijms232012444_

Round 1
Reviewer 1 Report
In the MS entitled “Modification of Regulatory T Cell Epitopes Promotes Effector 2 T Cell Responses to Aspartyl/Asparaginyl β-Hydroxylase” by Wirsching et al., the Authors identified using bioinformatics four (4) sequences (epitopes) within Aspartate-beta-hydroxylase (ASPH) presumed to be capable of stimulating Treg responses. Since ASPH is overexpressed in many human cancers, including Hepatocellular Carcinoma (HCC), but of low expression in normal, non-transformed tissues this membrane-bound enzyme has been held, since a few years ago, as a potential target for anti-cancer vaccination or else for immunotherapy. Indeed, epitope-specific and HLA-restricted, CD4+ and CD8+/CTL-mediated immune responses against ASPH have been reported by others (duly referenced by the Authors).
The caveat is that challenging with ASPH may also trigger a concurrent Treg-dependent immunosuppressive response, potentially precluding the development of efficacious vaccines, hence the relevance of the focus of the current piece of research.
In brief, in 9 healthy donors the Authors have, 1) identified four 9mer sequences with the potential to elicit a Treg response, 2) expressed a recombinant ASPH wt construct and a ASPH-modified (ASPHmod) construct in which the four Treg-specific sequences have been altered, 3) co-cultured ex-vivo circulating T cells with induced dendritic cells (DCs) loaded with either purified ASPH wt or ASPHmod proteins (or no antigen, controls), and 4) found that ASPHmod induced higher frequencies and higher proliferation rates of CD4+ Th1 cells (TNF-alfa, IFN-gamma positive) and CD8+ IFN-gamma positive cells, as well as higher cytokine production than ASPH wt; in addition, 5) challenging with ASPHmod led to lower frequencies of Tregs than a ASPH wt challenge. They concluded on the relevance of eliminating Treg-specific tolerogenic epitopes in developing ASPH-targeted vaccines in the future.
Although potentially interesting, this study faces some deficiencies in experimental design that might be improved.
MAJOR REMARKS:
The Reviewer is not asking for the more definitive proof that the four epitopes presumed to be Treg-specific are indeed MHC-bound, neither for the clonal characterization of the expanded Treg populations after stimulation by either ASPH wt or ASPHmod. However:
1) To improve the data, induced DCs should be challenged also with ASPHmod plus the Treg-specific peptides, either isolated or in combinations. This add-back experiments should revert (at least partially) the ASPHmod phenotype (low Tregs, high CD4+ and CD8+ Tcells) to the ASPH wt phenotype, thus providing added accuracy to the conclusions.
2) To further emphasize the putative role of Tregs in counteracting the expansion of CD4+ and CD8+ T cells, Tregs should also be depleted (eg, CD25 immuno-depletion) prior to incubation with DCs in some experiments. The additional use of drugs targeting Foxp3 could be wellcome (not as an alternative to Treg depletion)
MINOR REMARKS:
1) The identity of the ASPH wt and ASPHmod constructs should be verified by sequencing; this is not stated in Materials and Methods.
2) Although CD154 is considered a reliable marker of CD4+ T cell activation, CD137 might provide a better marker for detection of CD8+ T cell activation.
3) Although overall the text reads well the piece of text between lines 146-150 looks confusing!
Author Response
Dear Sir or Madam,
Please see the attachment for the point-by-point response.
Best regards,
Sebastian Wirsching

Reviewer 2 Report
The manuscript shows relevant data concerning the choice of antigens for the improvement of the efficacy of vaccine candidates. The results are consistent, and the conclusions are supported by the results. Minor revisions to the Materials and Methods and Results section are required. Please find the comments below:
In the YY axis (Fig 3 C and Fig 3D). Please clarify if these cells refer to CD4+ cells or CD3+ cells
Fig. 3C and Fig 3D represent the same set of results. The graphs are redundant. Suggestion: Put fig 3D as supplementary material
Fig 4C and 4D. Please clarify YY axis title refers to CD4+ cells or CD3+ cells
Figure 5 legend. ns is not represented in the figure why do you refer it in the legend?
line 146-151. This sentence is rather long and confusing. Please clarify what do you mean by ....." separate from ICCS sensitized T cells..... . The authors should rephrase the sentence and put some of this information in Materials and Methods section.
Line 152-154. How was the secretion of cytokines quantified? Please clarify.
The results on T cell proliferation would benefit from the inclusion of the flow cytometry analysis as supplementary material (Please add)
Line 246. Please refer that this result is not shown.
line 346 correct spelling error replace word "supernates" by supernatants
Author Response

(The authors gave the same response as above.)

Round 2
Reviewer 1 Report
The Authors have fairly and diligently responded to the major questions
This is a really improved version of the initial MS